# Structure Modification, Evolution, and Compositional Changes of Highly Conductive La:BaSnO_3_ Thin Films Annealed in Vacuum and Air Atmosphere

**DOI:** 10.3390/nano12142408

**Published:** 2022-07-14

**Authors:** Tomas Murauskas, Virgaudas Kubilius, Rimantas Raudonis, Martynas Skapas, Valentina Plausinaitiene

**Affiliations:** 1Institute of Chemistry, Faculty of Chemistry and Geosciences, Vilnius University, Naugarduko Str. 24, LT-03225 Vilnius, Lithuania; virgaudas.kubilius@chf.vu.lt (V.K.); rimantas.raudonis@chgf.vu.lt (R.R.); 2Center for Physical Science and Technology, Saulėtekio Av. 3, LT-10259 Vilnius, Lithuania; martynas.skapas@ftmc.lt

**Keywords:** chemical vapor decomposition, nonstoichiometry, perovskite oxide thin films, La:BaSnO_3_, annealing, surface morphology, decomposition, XPS

## Abstract

Perovskite-type La:BaSnO_3_ (LBSO) has been drawing considerable attention due to its high electron mobility and optical transparency. Its thin film electrical properties, however, remain inferior to those of single crystals. This work investigates the thermal post-treatment process of films deposited using the metalorganic chemical vapor deposition method to improve the electrical properties of different stoichiometry films, and demonstrates the modification of thin film’s structural properties using short and excessive annealing durations in vacuum conditions. Using vacuum post-treatment, we demonstrate the improvement of electrical properties in Ba-rich, near-stoichiometric, and Sn-rich samples with a maximum electron mobility of 116 cm^2^V^−1^s^−1^ at r.t. However, the improvement of electrical properties causes surface morphology and internal structural changes, which depend on thin film composition. At temperatures of 900 °C–1400 °C the volatile nature of LBSO constituting elements is described, which reveals possible deterioration mechanisms of thin LBSO air. At higher than 1200 °C, LBSO film’s decomposition rate increases exponentially. Thin film structure evolution and previously unreported decomposition is demonstrated by Ba and La diffusion to the substrate, and by evaporation of SnO-SnO_x_ species.

## 1. Introduction

Wide band-gap perovskite semiconductors attracted a great deal of research effort during the last decade [1]. La:BaSnO_3_ (LBSO), a single-crystal material, was one of the most promising candidates. Transparent in the visible spectrum range and demonstrating an unparalleled electron mobility of 320 cm^2^V^−1^s^−1^ (300 K) with a carrier concentration of ~10^19^ cm^−3^, LBSO could potentially overcome the carrier mobility limitations of other transparent oxide semiconductors. LBSO’s carrier mobility and low concentration result in high electrical conductivity, exceeding 10^4^ S cm^−1^ at room temperature, which could lead to applications in modern technology devices in optoelectronics such as touch-sensitive displays, thermally stable capacitors, gas sensors, etc. [2,3]. Its extraordinary single crystal properties led many researchers to fabricate thin films of LBSO. However, the determined carrier mobility of LBSO thin films was substantially lower (as a result of the formation of structural and point defects) than that of single crystals [1]. In recent years, the causes of many of these defects were identified, and possible methods have been proposed to increase the conductivity of thin films.

The first most notable defect responsible for LBSO’s inferior electrical properties was induced dislocations via film–substrate lattice mismatch. Therefore, various substrates with similar lattice parameters, such as DyScO_3_ and PrScO_3_, were used [4,5]. In both cases, using advanced and slow growth molecular beam epitaxy methods, electron mobility of 150 cm^2^V^−1^s^−1^ and 183 cm^2^V^−1^s^−1^ were achieved. Buffer layers were proposed as another option to address inferior electrical properties [6]. The most notable increase in film quality should have been achieved using homoepitaxial thin film growth on BaSnO_3_ (BSO) substrates; however, electrical properties remained considerably lower compared to single crystals [7].

Additional attempts using such remedies for LBSO deposition in scale would possibly be more costly than suitable post-treatment processes, which are usually employed to enhance thin film properties after deposition. Therefore, various thermal post-treatment options were investigated by many researchers. This did not lead to a radical increase in thin film properties. However, the electrical properties of the films were improved in the temperature range of 500–900 °C [8,9,10,11]. Thin films annealed in a vacuum showed an increase in carrier mobility and carrier concentration. However, the initial mobility of the reported films was low, ranging from 5–25 cm^2^V^−1^s^−1^. The cause of the mobility increase in LBSO films was generally considered to be oxygen vacancies acting as charge carrier donors. Other key factors, such as annealing durations ranging from 30 min to 8 h, were also investigated, but no significant improvements were observed. This work presents a more detailed annealing duration study and the resulting structural alterations, showing how thin film’s composition and structure change during short annealing times as well as excessively long annealing times. Additionally, different perovskite-type films were reported to undergo a surface reconstruction when treated thermally. For instance, SrTiO_3_ (STO) and (Ba, Sr, Ca):LaMnO_3_ underwent segregation of containing dopants or A-site cation oxides on the film surface depending on the annealing temperature, which demonstrates the importance of annealing temperature [12,13]. For this reason, considerable attention was given to surface chemistry. To clarify and expand the knowledge of potential surface changes, as well as film deterioration, annealing in significantly higher temperatures was investigated.

In addition, LBSO composition has been discussed as a crucial factor that determines the electrical properties of doped or undoped BSO films. Nonstoichiometry is an important factor; some deposition methods make it difficult to control thin film composition and might produce thin films with different elemental compositions and, thus, different electrical properties. Here a set of different composition samples were employed to reveal the different behaviors of nonstoichiometric LBSO films. The most common STO substrate was used, with various initial LBSO electron mobility, to examine near stoichiometric, Ba-rich, and Sn-rich film properties during the annealing process. Furthermore, this study encompassed several atmospheric conditions to investigate the influence of the annealing environment. Regarding all experimental factors, this work describes sufficient improvement of LBSO electrical properties, the volatile nature of several barium and tin species, and shows the possible reconstruction and degradation of the stannate phase.

## 2. Materials and Methods

Thin lanthanum-doped barium stannate films of ~200 nm were deposited using metalorganic chemical vapor deposition with a pulsed injection system, also known as PI-MOCVD [14]. Following metalorganic barium, tin and lanthanum precursors were used: Ba(thd)_2_, Sn(thd)_2,_ and La(thd)_3_ (thd—2,2,6,6-tetramethyl-3,5-heptandionate).

Single solutions of both Ba(thd)_2_ and La(thd)_3_ were used, while a Sn(thd)_2_ solution was prepared separately. The total concentration of all precursors in 1,2-dimethoxyethane (DME) was maintained at 0.019 M. To control thin films’ stoichiometry, the molar ratio between (Ba + La) and Sn was adjusted. Films were doped with lanthanum by mixing La(thd)_3_ and Ba(thd)_2_ at a molar ratio of (2.5% La/97.5% Ba). The precursor solutions were injected into the MOCVD system’s evaporators, heated to 200 °C. An argon and oxygen gas mixture of ratio 9:1 was used to transfer the metalorganic vapor to the crystalline (001) SrTiO_3_ substrates. The substrate was kept at 850 °C. The total pressure in the reactor chamber was 10 Torr, with a partial O_2_ pressure of 1 Torr. The obtained samples were used for the annealing experiments.

The first type of sample annealing was conducted using a tube furnace (MTI OTF-1200X-4-RTP, MTI) in 10^−1^ torr vacuum conditions at a temperature of 900 °C. The temperature was increased at a rate of 60 °C/s and decreased at 60 °C/min. The samples were annealed for different amounts of time ranging from 15 min to 36 h in progressively longer durations. The second type of annealing experiment was conducted using an annealing furnace in the air at atmospheric pressure for 8 h at temperatures of 1000, 1100, 1200, 1300, and 1400 °C. Furnace temperatures were increased at a rate of 5 °C/min and decreased at 60 °C/min. All samples were immediately investigated using X-ray photoelectron spectroscopy (XPS). The XPS analyses were performed using a Kratos Axis Supra spectrometer with a monochromatic Al K_α_ source (25 mA, 15 kV) probing the surface of the sample to a depth of 5–7 nm. The instrument work function was calibrated to give a binding energy (BE) of 83.96 eV for the Au 4f_7/2_ line for metallic gold, and the spectrometer dispersion was adjusted to give a BE of 932.62 eV for the Cu 2p_3/2_ line of metallic copper. The Kratos charge neutralizer system was used on all specimens. Survey scan analyses were conducted with an analysis area of 300 × 700 μm and a pass energy of 160 eV. High-resolution analyses were conducted with an analysis area of 300 × 700 μm and a pass energy of 20 eV. Spectra were charge corrected to the mainline of the carbon 1s spectrum (adventitious carbon) set to 284.8 eV. Sample etching was accomplished using a beam of monoatomic Ar^+^ ions of 5 keV and an etch area of 2 mm^2^. Spectra were analyzed using CasaXPS software (version 2.3.23rev1.1R). Structural high-resolution transmission electron microscopy (TEM) measurements of films formed on STO substrates were taken using FEI Tecnai G2 F20 X-TWIN TEM operating at 200 kV. A FEI Helios Nanolab 650 dual beam microscope equipped with an Omniprobe manipulator was used to prepare the cross-sectional TEM specimens. TEM lamella thickness was determined to be 60 nm using convergent beam electron diffraction pattern analysis.

The surface morphology and microstructure of the LBSO films were investigated using a scanning electron microscope (SEM SU70; Hitachi, Tokyo, Japan) operating at 10 kV. X-ray energy dispersion spectroscopy (EDX) was employed to determine elemental composition (SEM TM3000; Hitachi, Tokyo, Japan). Composition in this work is expressed as a molar Sn/Ba ratio, assuming low La doping. The crystalline structure and films’ crystal orientation were studied using an X-ray diffractometer with a Cu K_α_ radiation source (D8 Advance; Bruker, Berlin, Germany). The van der Pauw four-probe method and Hall measurements in a magnetic field of 0.6 T were used to measure electric resistance, charge carrier concentration, and density.

## 3. Results and Discussions

### 3.1. Time-Dependent LBSO Film Annealing in Vacuum

Thermal processing of LBSO with different stoichiometry was the focus of this work. Therefore, careful consideration of initial experiment conditions were required to determine structural modification or certain cation redistribution in the films. Lanthanum-doped barium stannate’s (LBSO) electrical properties are known to depend on the substrate lattice parameter. Film–substrate mismatch was shown to induce structural and point defects at the interface, which propagate further to the film [1,5]. To minimize negative effects on thin structural and, hence, electrical properties, STO (100) substrate of similar lattice parameters were used in this study. Additionally, a slight deviation from stoichiometric composition to Ba-rich or Sn-rich composition in our previous work showed a decrease in electron mobilities which were strongly dependent on the concentration of point defects [15]. To account for possible contributions of different defect concentrations of the films, a sample set of different compositions was chosen. La doping was maintained constant at a concentration of 2.4 ± 0.3%_atomic_ relative to Ba. The Sn/Ba ratios and electron mobility of the initial samples were: 0.93 (7.0 cm^2^V^−1^s^−1^), 0.96 (45 cm^2^V^−1^s^−1^), 0.98 (105 cm^2^V^−1^s^−1^), 1.03 (18 cm^2^V^−1^s^−1^), 1.06 (8 cm^2^V^−1^s^−1^), and 1.09 (6 cm^2^V^−1^s^−1^). All samples were deposited at 850 °C. LBSO was stable during the synthesis temperature of 850 °C and able to withstand this temperature for more than 12 h. To investigate and observe changes in LBSO films in controlled conditions, a slightly higher temperature of 900 °C was used. Vacuum sample annealing was performed to improve the film’s electrical properties, such as mobility and specific resistance. To observe a more significant change, samples were annealed in consecutively longer durations from 15 min to the longest duration of 36 h.

XRD patterns of all the samples were recorded after each cycle. Samples with Sn/Ba = 0.98 composition are provided as an example in Figure 1a. The results showed that after annealing for 36 h thin films remained epitaxial and of a single phase. The calculated lattice parameter was 4.117 ± 0.001 Å. No significant structural or change in out-of-plane lattice parameters, *a*, were observed, even after 36 h in vacuum conditions.

Improvement of electrical properties was observed after annealing in a vacuum. Electrical measurements showed that film conductive properties were slightly improved by shorter annealing times. After 30 min, electron mobility increased in the most conductive sample of Sn/Ba = 0.98 from 103 to 116 cm^2^V^−1^s^−1^, as seen in Figure 1b. A relatively more significant increase was observed for samples with nonstoichiometric composition annealed for 30 min. The change in mobility with increasing annealing time was very similar in the regions of Sn-rich and Ba-rich samples. Sn-rich samples’ (Sn/Ba = 1.03) mobility increased from 18 to 48 cm^2^V^−1^s^−1^, while Ba-rich samples’ (Sn/Ba = 0.96) mobility increased from 46 to 65 cm^2^V^−1^s^−1^. In addition, in the case of a larger deviation from stoichiometry, we observed a tendency that increases in electron mobility were relatively higher after annealing and before annealing. The resulting values, however, remained relatively low. In comparison, the carrier concentration showed a dependence only on the initial carrier concentration. According to other studies, oxygen vacancies could form and act as additional carrier donors if ionized. Vacuum annealing at 900 °C produced no significant increase in carrier concentration. This suggested that vacancies did not ionize or were compensated when samples were exposed to ambient conditions, as seen in Figure 1c [16].

Figure 1d shows the specific conductivity of as-deposited as annealed thin films. The tendencies of conductivity are closely related to mobility and follow similar behavior. Conductivity increases for most samples annealed up to 30 min. Both conductivity and mobility dependencies are possibly related to structural factors. In other studies, annealing at high temperatures in N_2_ or H_2_ was shown to induce lateral grain growth [17]. Lateral grain growth significantly increased free carrier propagation length and, thus, electron mobility [8,18,19,20]. As post-annealing carrier concentration remained similar, a small improvement in film conductivity was most likely related to an increase in grain size or changes in composition. High deposition temperatures and a slow cooling process, in this case, might have acted as in situ annealing.

XRD and electrical measurements showed the average properties of the film, but changes to the very surface of the samples are not reflected in those measurements. Therefore, sample morphology was investigated using scanning electron microscopy after each annealing. Similar to our previous study [21], MOCVD-produced samples had rough surfaces when sample stoichiometry was shifted to Ba-rich or Sn-rich composition. Comparing all these samples, a similar behavior was observed. Short annealing, ranging from 15 to 30 min, showed no significant change in sample morphology, as shown in Figure 2. In contrast, a more distinct difference was noticed after more than 4 h, with the most significant change seen at 36 h (marked in red box). This annealing roughened the sample surface. The emerging crystallites of the film could be observed. This film surface reconstruction suggested that a part of the film’s mass was removed from the surface, or that the changes were related to certain recrystallization processes.

To better understand surface changes seen in the SEM images, X-ray photoelectron spectroscopy (XPS) and EDX were used to investigate the samples. XPS extracts information from 5–10 nm of the sample, while EDX probes the full depth of the layer.

EDX results showed that with increasing annealing duration, the overall Sn/Ba ratio decreased. The most noticeable decrease was measured for the films containing an initial excessive amount of tin (Sn/Ba = 1.06, 1.04, and 1.02), as the Sn/Ba ratio dropped below 1.00. In the case of Ba-rich samples, composition shifts were relatively small. The faster decrease in tin content in Sn-rich films might be related to the evaporation of tin oxide species. The evaporation of SnO_2_ oxide is unlikely due to its high melting and sublimation temperatures (~1600 °C and <1800 °C). In contrast, its reduced forms’ (SnO and SnO_x_) melting points are lower, at approximately 1400 °C [22]. This suggests two possible options; either the Sn surplus in the films is composed of the reduced form of tin oxide SnO-SnO_x_, or SnO_2_ species are gradually reduced in vacuum conditions and sublimate as reduced oxides. EDX measurements demonstrated that in these aggressive annealing conditions, LBSO layers remained relatively stable, even after 36 h, and the Sn/Ba ratio did not decrease below 0.91. However, the Sn/Ba ratio highly depended on the initial sample composition and was most significant for the Sn-rich samples. The actual case is well described in further XPS results.

In comparison, XPS measurement of the same sample surface showed a significant difference between surface and bulk compositions (EDX). In XPS analysis, the Sn/Ba ratio was calculated from high-resolution spectra by fitting spectra using Shirley background and GL(30) peakshapes. As-deposited samples with a higher bulk Sn/Ba ratio showed a significantly higher tin concentration on the sample surface: Sn/Ba EDX = 1.09 vs. Sn/Ba XPS = 2.48. After the first 15 min of annealing, a near-stoichiometric surface composition (Sn/Ba~1) was achieved. Longer annealing times drastically stripped the sample surface of tin content, reducing the Sn/Ba ratio to ~0.25. This was further investigated by depth profiling the annealed films, which did not show significant surface accumulation of barium. This suggested that the rapid decrease in the Sn/Ba ratio was most likely the result of Sn oxide species volatility. Therefore, detailed qualitative high-resolution XPS spectra analysis was performed.

XPS qualitative measurements were also used to investigate the valence states of ions and the surface composition. To avoid damaging the samples and possibly altering the surface, the usual Ar^+^ etching was not performed. All binding energies were referenced to the C 1s peak defined at 284.8 eV. The oxidation state of La ions was analyzed first; its signals were of low resolution due to small doping concentrations of 2.4 ± 0.3%. However, typical two doublet peaks were detected in each sample (La 3d_3/2_ and La 3d_5/2_) close to those expected for a La^3+^ ion in an oxide environment [23], and no noticeable differences were observed in the doublets. Following lanthanum analysis, the Sn 3d spectra were studied (Figure 3c). The double spectral lines for Sn 3d_3/2_ and 3d_5/2_ were detected at 495.9 eV and 494.3 eV, respectively. No metallic Sn^0^ signals were detected. Spin–orbit splitting of 8.4 eV was obtained, which is attributed to the Sn^4+^ chemical species [24]. However, this value is also typical for other tin oxidation states [25]. The more useful parameter is the Auger parameter (AP), which was demonstrated to significantly differ, depending on tin oxidation states. The tin Auger parameter, which is the sum of Sn 3d_5/2_ binding energy and Sn M_4_N_45_N_45_ kinetic energy, varies from 922.3 eV for tin metal to 919.2 eV for SnO_2_ according to the NIST database and other sources [14,25,26,27]. In our samples, the AP was measured to be approximately 919.3 eV. Thus, it is more likely that tin in all the samples remained in the Sn^4+^ state. Additionally, spin–orbit splitting gradually increased to 8.6 eV as the annealing time increased. Tin’s peak FWHM values also increased from 1.35 eV to 1.65 eV. This indicated that the measured spin–orbit splitting increase was possibly related to an additional doublet component in the Sn 3d signal, attributed to separate tin oxide species.

Similar behavior was observed in Ba 3d spectra. Ba 3d_3/2_ and 3d_5/2_ were detected at 779.2 eV and 794.3 eV, respectively, with spin–orbit splitting of 15.1 eV and FWHM (Ba 3d_5/2_) of 1.73 eV. Similar to Sn spectra, the spin–orbit splitting and FWHM values increased to 15.4 eV and 1.94 eV, respectively, indicating that barium spectra were composed of at least two contributions. As in other perovskite materials, such as SrTiO_3_, segregation of BaO might have been expected. Such a process would have resulted in an increase in Ba on the surface changes [28]. However, O 1s spectra revealed different chemical species and showed that both Ba and Sn spectra might be the result of increasing oxygen deficiency and the difference in cation chemical surroundings, as well as the experimental conditions [29].

O 1s spectra provided in Figure 3d demonstrated the lattice oxygen of LBSO at ~530 eV and an increase in additional components at higher binding energies of 531–532 eV. O 1s spectra components at B.E. > 531 eV were related to either adventitious carbon–oxygen or adsorbed water. The calculated carbon-related oxygen amounted to a maximum of 5% of the total oxygen and was ruled out as the origin of higher energy signals. Additionally, C 1s high resolution spectra showed no features at binding energies higher than 288 eV, which could correspond to carbonate signals. Thus, as-deposited and layers annealed for 15 min showed dissociatively adsorbed water molecule signals. With increased annealing time, the sample surface was most likely depleted of oxygen, which should have increased the oxygen vacancy concentration and carrier density. The carrier density, however, did not increase, and an increase in additional oxygen species suggested that formed vacancies were compensated. As all the experiments took place ex situ, the reduced surface most likely reacted with moisture in the air, forming chemically bound hydroxyl groups in Ba(OH)_2_ on the sample surface. The increasing surface hydroxide amount in Ba(OH)_2_ was most likely the origin of energy shifts and the increase in the aforementioned FWHM values of the Ba 3d and Sn 3d spectra. The existence of chemosorbed-OH groups was tested by heating samples in ultra-high vacuum conditions of an XPS instrument up to 850 °C for 10 min. If thin films had adsorbed water to repair surface damage by forming -OH groups, the oxygen species would have been removed at high temperatures as Ba(OH)_2_ gas [30,31,32,33]. Ultra-high vacuum annealing resulted in a decrease in the FWHM values of Ba 3s and Sn 3d signals, as well as an increase in the O 1s lattice signal, indicating the removal of both Ba and Sn volatile species. These results suggested the existence of volatile surface Ba(OH)_2_ and different species of SnO-SnO_x_ surface compositions [34]. This was also confirmed using XPS depth profiles that demonstrated that only the surface was affected by the annealing. As a result of selective ion etching, XPS depth profiling was not an accurate representation of the chemical composition and is not presented here. However, it did provide a very similar composition profile between annealed and as-deposited samples, showing that the bulk volume of the films remained similar.

XPS data revealed that the sample surface was altered during the annealing process and that additional chemical species were formed on the surface. Longer annealing durations resulted in higher amounts of different Ba and Sn phases. Both Ba(OH)_2_ and SnO-SnO_x_ are insulators and did not affect the carrier mobility or concentration of the film. These results suggested that structural changes in the bulk of the film were the most likely factor.

For a detailed investigation of structural changes, TEM measurements were performed. Figure 4a,b show a cross-section view of thin films. Thin films can be characterized by columnar growth with apparent crystallite boundaries, marked by blue arrows. Both films contain sharp curved lines running along [001] direction which are neither pure edge dislocations nor threading dislocations, and are most likely generated by grain twist along [001] direction. This specific view is often observed in LBSO films [6,35]. In both cases, the film–substrate interface was seen as a dark contrast area, resulting from strain misfit dislocation sites due to film–substrate lattice mismatch. Additionally, the deposited sample showed a stacking defect, with atomic planes inclined to each other by small angles (ripple pattern in the yellow box), as reported by Yun et al. This type of defect was not present in the annealed film, probably indicating the annihilation of such defect through annealing.

Additionally, the annealed samples showed an increase in sharp diagonal features (marked by yellow arrows). These features might be a product of ongoing recrystallization rather than the formation of threading dislocations. This is supported by the appearance of voids in the film–substrate interface (marked by red circles). The formation of voids and a significant decrease in the Sn/Ba ratio indicates that tin content is removed from the grain boundaries of the bulk, leaving voids in the film. After 36 h at 900 °C, a secondary grain growth along the most packed [111] direction was observed [36]. A similar recrystallization process was demonstrated by other authors [37]. This was also supported by the high-resolution images in Figure 4c,d. While the deposited film demonstrated a high density of edge dislocation and stacking faults, no such features were observed in the annealed samples. This might suggest that an increase in the before-mentioned electron mobility might be related to a faster decrease in tin content and increase in grain boundaries order. TEM analysis showed that the annealing process increased the crystalline order of the films and resulted in a decrease in various stacking defects; it also formed voids in the bulk of the film.

### 3.2. LBSO Film High-Temperature Annealing in Air

In this study, vacuum annealing showed noticeable compositional changes in the film surface and structural changes in bulk due to the removal of Ba(OH)_2_ and SnO-SnOx species. The surface composition was most noticeable in a vacuum. In comparison, D. Yoon et al., who used different atmospheric conditions (Ar, O_2_, and wet H_2_) at 1000 °C for 1 h, observed the formation of a terraced surface, or the formation of BaO_x_ islands. No significant composition changes were reported after annealing in the Ar atmosphere. Comparing these results, it is noted that the atmosphere plays a key role in the annealing process in LBSO and determines its chemical composition, which is also closely related to the annealing temperature. However, in the annealing studies we found, the stability and chemical changes in LBSO were not investigated at higher temperatures. Therefore, to investigate the thermal stability threshold, films were annealed at consecutively higher temperatures for 8 h, at 1000 °C, 1100 °C, 1200 °C, 1300 °C, and 1400 °C in atmospheric pressure. Annealing at such temperatures should not affect the substrate due to its thermal stability up to 1640 °C [38]. Electrical properties were investigated first, but all samples showed high resistance values due to deviation in composition lower than Sn/Ba = 0.8. No mobility or carrier concentrations could be determined.

In Figure 5a–e, the SEM images of annealed thin film surfaces are provided. Thin film surfaces before the annealing were similar to the surface given in the Figure 1 (Sn/Ba = 0.93) sample before annealing. After annealing, a distinct difference was determined. At 1000–1200 °C the layers became more fragmented and discernible ridges and symmetric crystallites formed on the sample surface. For longer annealing times, triangle-shaped surfaces appeared on the film surface, as shown in Figure 5f. A mixed-type SEM imaging of backscattering and secondary electron detectors enhanced the shape features and provided a qualitative analysis of the surface [39]. However, it was determined that these features were most likely of the same phase and formed due to recrystallization; comparing topographical images of SEM and emerging diagonal crystallite boundaries in TEM images in Figure 4b, such origin of these shapes was most likely. At 1200 °C additional reflections were detected, suggesting the emergence of additional phases. As the triangle-shaped features were observed in the range of 1000–1200 °C, detected XRD signals were most likely not related to the observed additional phase.

After annealing at 1300 °C, a highly textured film surface with ridges was observed (Figure 5e). XRD data showed several changes in the film. Firstly, additional reflections were observed at 28.21°, 39.14°, and 66.78° (Figure 5g) which could not be attributed to BaO or SnO_2_. Secondly, substrate peaks were broader and showed a shouldering peak at smaller angles to STO (001) reflections (marked by asterisks in Figure 5g). This indicated that an additional phase similar to the substrate had formed. This was possible if the film reacted with the substrate; exactly this was indicated by the XRD and EDX data shown in Figure 5h. The determined Sn/Ba ratio decreased significantly to 0.58. At such a ratio, an additional LBSO degradation phase would have formed. However, the LBSO reflections not only remained at the same angles, but showed an increase in crystallinity indicated by the lower half-widths of LBSO (001). The remaining Ba did not form the BaO oxide phase. It is most likely that Ba and La diffused into the interface of the substrate. This was indicated by the XRD, as additional reflections also matched several mixed oxide phases, such as BaTiO_3_, (Ba_0.59_Sr_0.408_)TiO_3_, Ba_2_TiO_4_, and Ba_0.3_Sr_0.7_TiO_3_.

BaTiO_3_ species were excluded based on phase diagrams showing formation of a cubic BaTiO_3_-liquid Ba_2_Ti_2_O_7_ phase [40]. The remaining (Ba_0.59_Sr_0.408_)TiO_3_, BaO + Ba_2_TiO_4_, and Ba_0.3_Sr_0.7_TiO_3_ were most likely candidates regarding the excess Ba content. These surprising results were further clarified by investigating the sample annealed at 1400 °C.

The LBSO phase was not detected at 1400 °C, indicating total deterioration of the LBSO film. The sample surface highly resembled the surface of the substrate. In this case, XRD data showed formation of (Ba_1.5_Sr_0.5_)TiO_4_ and Ba_0.3_Sr_0.7_TiO_3_ phases and a STO signal. An XPS depth profile confirmed high atomic concentrations of Ba, La, Ti, and Sr as deep as ~50 nm into the substrate surface. Only trace amounts of tin were detected. These data reveal that in high temperatures above 1200 °C, Ba and La can diffuse into the similar structure of STO, forming intermediate compounds and incorporating the La dopant.

EDX data given in Figure 5h also indicated that annealing at temperatures higher than 1200 °C caused a significant decrease in tin content and change in composition. At these higher temperatures tin oxide evaporation was very significant. Tin species’ evaporation rate increased exponentially with the increase in temperature. At 1400 °C only a trace amount of Sn was detected, resulting in the ridge-like morphology and the formation of intermediate LBSO-STO phases. Further studies are needed to investigate whether the deterioration of LBSO at such temperatures is dependent on the nature of the substrate.

## 4. Conclusions

The MOCVD method was successfully used to deposit La:BaSnO_3_ samples with near stoichiometric composition with the desired deviation of the Sn/Ba ratio. The highest increase in mobility was observed when the samples were annealed for 30 min at 900 °C; however, the increase in mobility in nonstoichiometric samples was relatively low. Extended vacuum annealing for 36 h showed that LBSO films were relatively stable, even though it induced the formation of new grain boundaries and voids in the bulk of the film. Thermal processing the films at 900 °C affected the surface composition and reduced the Sn amount in the surface; it also formed hydroxide groups by compensating for the formed oxygen vacancies when films were exposed to air. In contrast, sample annealing in air at higher temperatures irreversibly altered film composition by depleting films of tin species, resulting in a decrease in measurable mobility. Temperatures higher than 1200 °C on the SrTiO_3_ substrate caused the diffusion of Ba and La to the substrate surface and formed additional phases with the substrate. It was determined that prolonged annealing in air at higher temperatures can cause a significant deterioration of La:BaSnO_3_ on SrTiO_3_ substrate.

## Figures and Tables

**Figure 1 nanomaterials-12-02408-f001:**
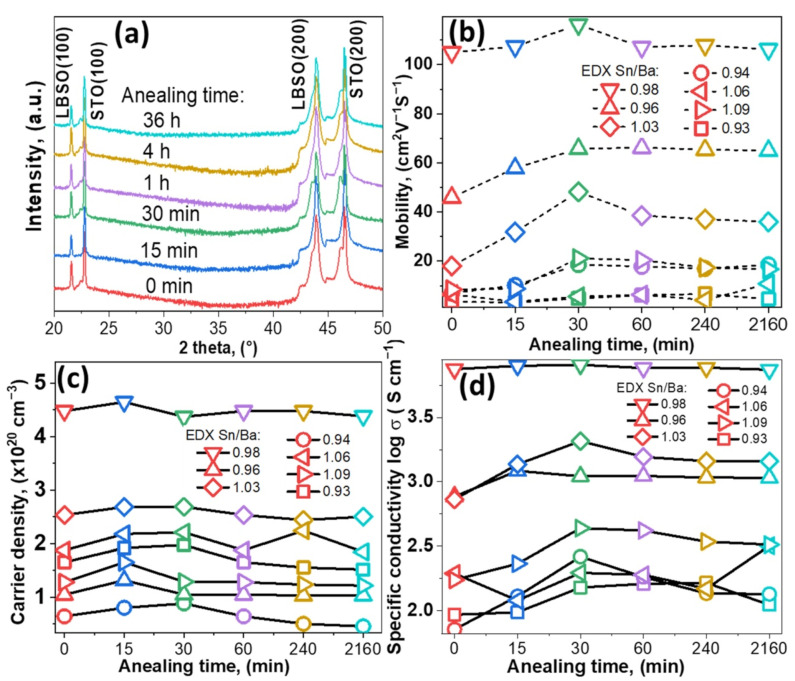
(**a**) XRD pattern of single-phase LBSO sample with Sn/Ba ratio of 0.98 annealed at 900 °C for different durations, ICDD PDF #00-042-1468. (**b**–**d**) Electron mobility, carrier density, and specific conductivity of different LBSO samples with their respective Sn/Ba ratios (0.93, 0.94, 0.96, 0.98, 1.03, 1.06, and 1.09) annealed for different durations.

**Figure 2 nanomaterials-12-02408-f002:**
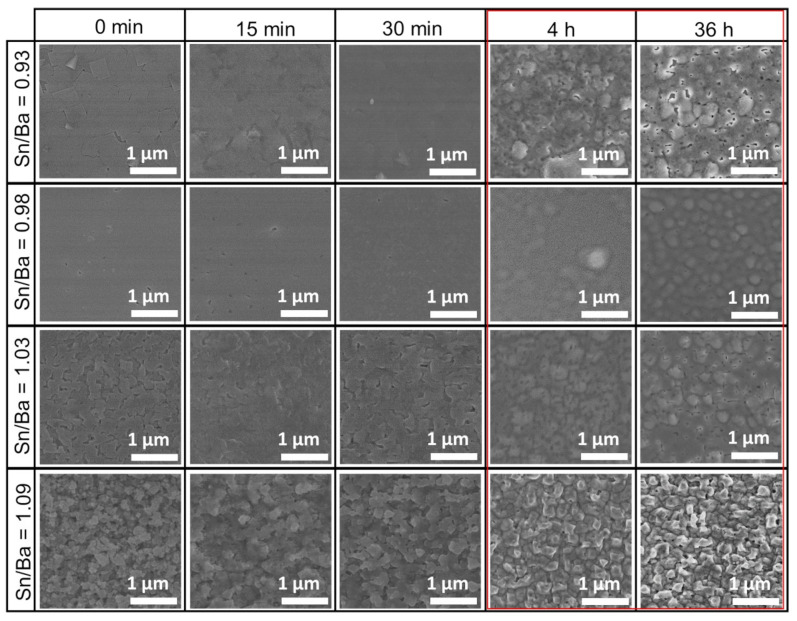
SEM morphology evolution of thin La-doped barium stannate films deposited on SrTiO_3_ (001) substrates. Samples were annealed at 900 °C for different durations. Each column represents durations of as-deposited, 15 min, 30 min, 4 h, and 36 h. Each row represents a different initial Sn/Ba molar ratio of the samples corresponding to 0.93, 0.98, 1.03, and 1.09. The red box marks the most significant changes in the LBSO films’ morphology.

**Figure 3 nanomaterials-12-02408-f003:**
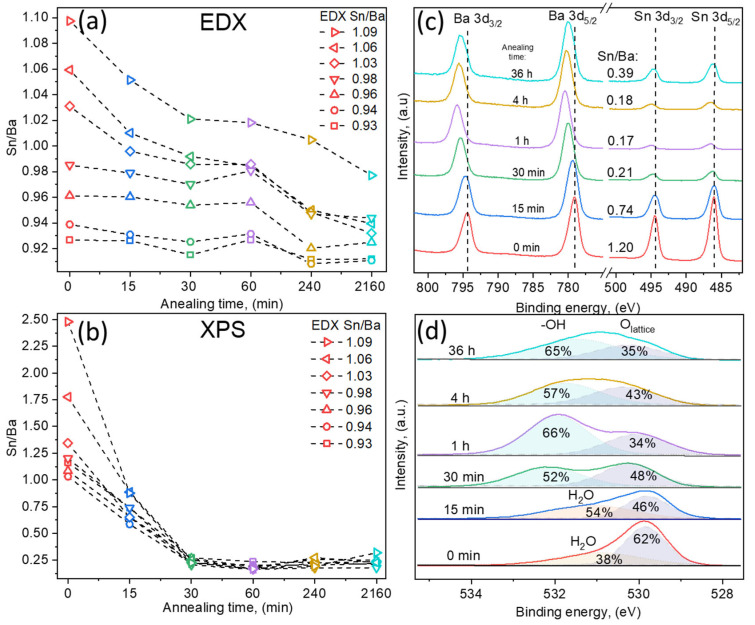
Composition dependency on annealing time of LBSO films with Sn/Ba ratios of 0.93, 0.94, 0.96, 0.98, 1.03, 1.06, and 1.09 as determined using EDX. Samples are named using the initial Sn/Ba ratio determined using EDX as a reference. (**a**) Bulk thin LBSO film Sn/Ba ratio dependency on annealing time measured using EDX. (**b**) Surface Sn/Ba ratio at different annealing times determined using X-ray photoelectron spectroscopy. (**c**) High-resolution Ba 3d and Sn 3d XPS spectra of LBSO film with an initial Sn/Ba ratio of 0.98. (**d**) High-resolution O 1s XPS spectra dependency on annealing time of LBSO sample with Sn/Ba ratio of 0.98.

**Figure 4 nanomaterials-12-02408-f004:**
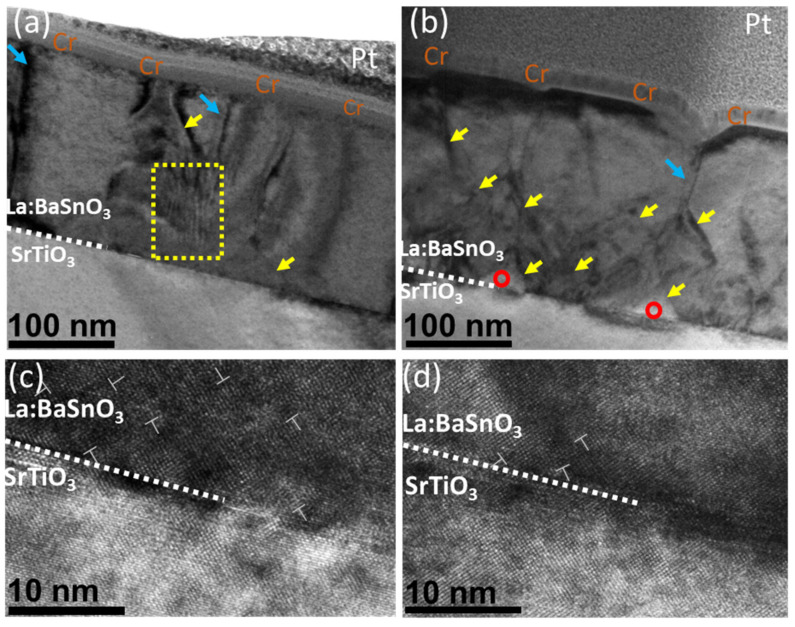
(**a**) Deposited and (**b**) annealed cross-sectional HAADF-STEM image of a La:BaSnO_3_ film grown on a SrTiO_3_ substrate. Blue arrows mark grain boundaries. Yellow arrows show threading dislocations. Red circles mark voids in the film. Yellow box Plan-view HAADF-STEM images of the (**c**) deposited and (**d**) annealed La:BaSnO_3_ films with two unique line defects (in the yellow boxes) and one edge dislocation (indicated by a dislocation symbol).

**Figure 5 nanomaterials-12-02408-f005:**
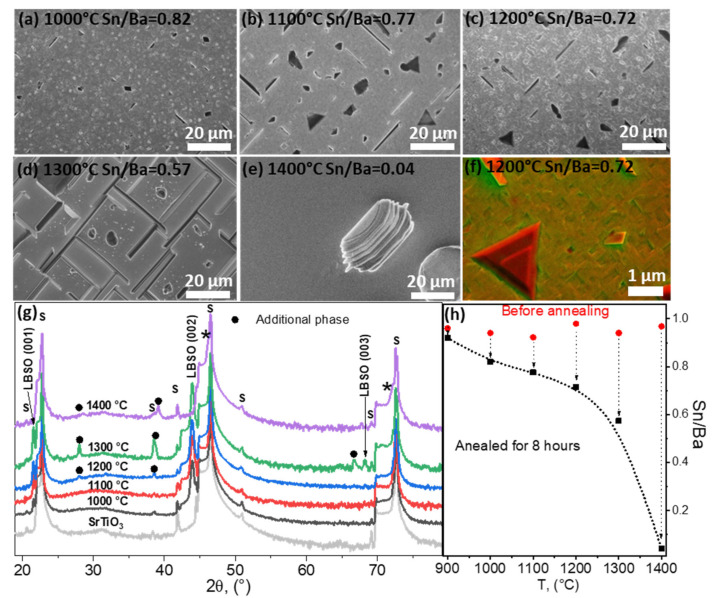
(**a**–**e**) SEM images of thin LBSO films annealed at different temperatures and their compositions after annealing determined using EDX. (**f**) SEM image using backscattered and secondary electron detectors simultaneously. (**g**) XRD parallel-beam scans of annealed thin La-doped BaSnO_3_ (ICDD PDF #04-018-6953) films deposited on (100) STO (ICDD PDF #00-005-0634) (marked by “s”). Black dots and asterisks (*****) mark possible additional phases: (Ba_0.30_Sr_0.70_)TiO_3_ (ICDD PDF #04-008-0742), (Ba_0.59_Sr_0.408_)TiO_3_ (ICDD PDF #01-070-3628), and Ba_0.97_La_0.3_SnO_3_ (ICDD PDF #04-018-6953). (**h**) EXD composition analysis of the as-deposited and 8 h annealed samples at different temperatures.

## Data Availability

Not applicable.

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
