# Peer review of "Structure Modification, Evolution, and Compositional Changes of Highly Conductive La:BaSnO3 Thin Films Annealed in Vacuum and Air Atmosphere"

_nanomaterials, 2022, doi:10.3390/nano12142408_

Round 1

Reviewer 1 Report

Very interesting paper concerning detailed study of La-doped BaSnO3 thin films. It can be accepted for publications after minor revision. Here come some comments. 1. La-doping should lead to the formation of A- or B-cation deficient perovskite phase due to electoneutrality principle. What kind of defeciency are you expecting? 2. It is rather common an enrichment of a surface of a  perovskite phase with A-cations. The formation of BaO  is hardy to expect due its very high chemical reactivity. However, one can expect the formation of very stable BaCO3  phase, especially if you started from C-containing precursors or a sample was subjected to air. Did you check the formation of BaCO3 phase at the surface of your samples? 3. Do you expect the formation of different RP phases like Ba2SnO4 or Ba3Sn2O7 at the surface as defects? Can you see such defects by HREM? 4. Please, make reference to ICDD database like ICDD PDF # 00-005-0634 and use this style everywhere in the text (there are 2 different styles in the text).

Reviewer 2 Report

I had a feeling that the paper was not organized well about the discussions.

In the title of the manuscript is "Structure modification, evolution, and compositional changes of highly conductive La:BaSnO3 thin films annealed in vacuum and air atmosphere".  The manuscript shows the results of mobility and carrier density. However, I can't find the data of conductivity.  Hence, please show the results of conductivity, and summarize the discussion with the data of conductivity.

Checking the data of mobility and carrier density, the best conductivity has been obtained at 0.98 (= Sn/Ba).  However, there is no comparison of the XRD data around 0.98 (= Sn/Ba), in order to compare the structure.  Moreover, in XPS results (Figure 3b), we can't see the results of 0.98 (= Sn/Ba).

Please revise the manuscript for the wider readers and for the future of scientifical society. 
